# Human Learning by Model Feedback:
# The Dynamics of Iterative Prompting with *Midjourney*

**Shachar Don-Yehiya**[‡]    **Leshem Choshen**[§]    **Omri Abend**[‡]

[‡]The Hebrew University of Jerusalem, [§]MIT

{first.last}@mail.huji.ac.il

## Abstract

Generating images with a Text-to-Image model often requires multiple trials, where human users iteratively update their prompt based on feedback, namely the output image. Taking inspiration from cognitive work on reference games and dialogue alignment, this paper analyzes the dynamics of the user prompts along such iterations. We compile a dataset of iterative interactions of human users with Midjourney.[1] Our analysis then reveals that prompts predictably converge toward specific traits along these iterations. We further study whether this convergence is due to human users, realizing they missed important details, or due to adaptation to the model's "preferences", producing better images for a specific language style. We show initial evidence that both possibilities are at play. The possibility that users adapt to the model's preference raises concerns about reusing user data for further training. The prompts may be biased towards the preferences of a specific model, rather than align with human intentions and natural manner of expression.

## 1 Introduction

Text-to-image models have become one of the most remarkable applications in the intersection of computer vision and natural language processing (Zhang et al., 2023). Their promise, to generate an image based on a natural language description, is challenging not only to the models, but to the human users as well. Generating images with the desired details requires proper textual prompts, which often take multiple turns, with the human user updating their prompt slightly based on the last image they received. We see each such interaction as a "thread", a sequence of prompts, and analyze the dynamics of user prompts along the interaction. We are not aware of any work that examines the dynamics of the prompts between iterations.

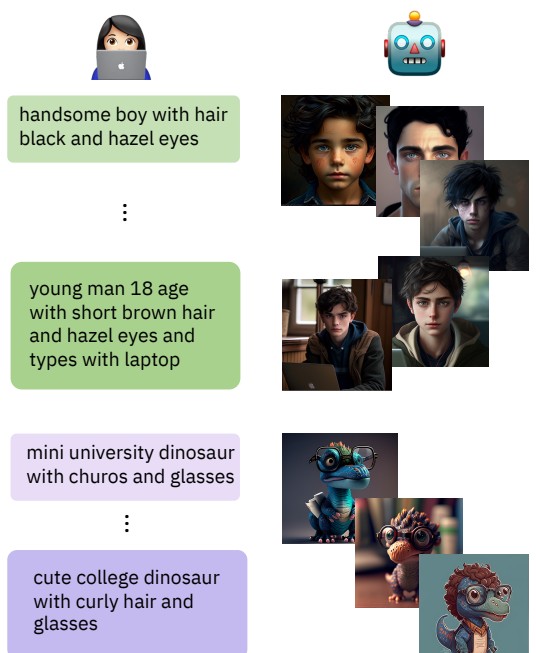

Figure 1: Threads examples. The user adjusts their prompts along the interaction. The first thread gets more concrete ("young man 18 age" instead of "boy") and longer ("types with laptop"). The second changes wording ("cute" instead of "mini", "college" instead of "university").

To study this question, we compile the Midjoureny dataset, scraped from the Midjoureny Discord server[2], containing prompts and their corresponding generated images and metadata, organized as $107,051$ interaction threads.

The language people use when they interact with each other changes over the course of the conversation (Delaney-Busch et al., 2019). Theoretical work suggests that learning mechanisms may allow interlocutors to dynamically adapt not only their vocabulary but their representations of meaning (Brennan and Clark, 1996; Pickering and Garrod, 2004, 2021). We hypothesize that also when in-

---

[1]Code and data are available in: https://github.com/
shachardon/Mid-Journey-to-alignment

[2]https://www.midjourney.com

teracting with Midjourney, where only the human user is able to adapt and the Midjourney model remains "frozen", we would see a systematic language change along the iterations.

Unlike the interaction with general assistant models (Köpf et al., 2023), which might include multiple topics and change along the conversation, the interaction thread of a text-to-image model contains attempts to generate one image, a single scene, with no major content change. This allows us to better recognize the change in the linguistic features rather than the content.

Our analysis reveals convergence patterns along the threads, i.e., during interactions humans adjust to shared features that bring them closer to their ideal image in terms of prompt length, perplexity, concreteness and more. Still, it is unclear whether these adjustments are due to humans adding missing details, or due to matching the **model preferences** – generating better images due to prompts in a language style that is easier for it to infer, thus encouraging users to adapt to it. We find evidence for both.

The second possibility, that users adapt to the model preferences, calls for caution regarding the subsequent use of human data from human-model interaction. For example, we could take the "successful" images that the human users presumably liked and requested a high resolution version of them ("upscale"), and use them with their matching prompts as a free human-feedback dataset (Bai et al., 2022). However, given that these prompts may be biased towards the model's preferences, training on them would create a model that has even more 'model-like' behaviour.

We hope that by releasing this first iterative prompting dataset, along with our findings regarding possible biases in the human data, we would encourage more work on human-model alignment and interaction.

## 2   Background

In a *repeated reference game* (Clark and Wilkes-Gibbs, 1986), pairs of players are presented with a set of images. On each iteration, one player (the director) is privately shown which one is the target image, and should produce a referring expression to help their partner (the matcher) to correctly select that image. At the end of the iteration, the director is given feedback about which image the matcher selected, and the matcher is given feedback about

the true target image. Empirical findings show a number of recurring behavioral trends in the reference game task. For example, descriptions are dramatically shortened across iterations (Krauss and Weinheimer, 1964), and the resulting labels are partner-specific (Wilkes-Gibbs and Clark, 1992; Brennan and Hanna, 2009).

We view the interaction thread as similar to a repeated reference game of the human user with the model. The human user directs Midjourney with textual prompts to generate (instead of select) the target image. Unlike the original game, only the human user is changing along the interaction based on the feedback (i.e., the image) they get from Midjourney. The Midjourney model is 'frozen', not able to adjust to the user feedback.

We hypothesize that also in our semi-reference game where only the human user is able to adapt we would see a language change along the iterations. We use similar methods to those used in recent works (Ji et al., 2022; Hawkins et al., 2019), in order to examine this change.

Prompt and image pairs, together with their meta-data about upscale requests, can be seen as a great source of data for *Reinforcement Learning from Human Feedback* (RLHF). In RLHF, non-expert human preferences are used to train the model to achieve better alignment (Christiano et al., 2017; Bai et al., 2022; Lee et al., 2023; Wu et al., 2023). We discuss in §9 the possible consequences of reusing the Midjourney data.

## 3   Data

In this section, we describe the choices and process of acquiring text-to-image interactions. We start by discussing the reasons to pick Midjourney data rather than other text-to-image data sources.

One reason to prefer Midjourney over open-source text-to-image models is its strong capabilities. Midjourney can handle complicated prompts, making the human-model interaction closer to a standard human-human interaction.

Another reason to prefer Midjourney is the availability of the prompts, images and meta-data on the Discord server. We construct the dataset by scraping user-generated prompts from the Midjourney Discord server. The server contains channels in which a user can type a prompt and arguments, and then the Midjourney bot would reply with 4 generated images, combined together into a grid. Then, if the user is satisfied with one of the 4 images, they

can send an "upscale" command to the bot, to get an upscaled version of the desired image.

We randomly choose one of the "newbies" channels, where both new and experienced users are experimenting with general domain prompts (in contrast to the "characters" channel for example). We collect $693,528$ prompts (From 23 January to 1 March 2023), together with their matching images and meta-data such as timestamps and user ids (which we anonymize).

In §F we repeat some of the experiments with data from Stable Diffusion (Rombach et al., 2021), concluding that our results can be extended to other models.

### 3.1 Data Cleaning

The Midjourney bot is capable of inferring not only textual prompts, but also reference images. Since we are interested in the linguistic side, we filter out prompts that contain images. We also limit ourselves to prompts in the English language, to allow a cleaner analysis.[3] We remove prompts with no text, or no matching generated image (due to technical problems). After cleaning, we remain with $169,620$ prompts.

The Midjourney bot can get as part of the input prompt some predefined parameters like the aspect ratio, chaos and more,[4] provided at the end of the prompt. We separate these parameters from the rest of the text, so in our analysis we will be looking at natural language sentences. In §C we repeat some of the experiments with prompts with the default parameters only (i.e., with no predefined parameters).

### 3.2 Data Statistics

The dataset contains prompts from $30,394$ different users, each has $5.58$ prompts on average with standard deviation $20.52$. $22,563$ users have more than one prompt, and $4008$ of them have more than 10 each.

### 3.3 Upscale

As mentioned, when a user is satisfied with one of the grid images, they can send an upscale command to obtain an upscaled version of it. We collect these commands, as an estimation to the satisfaction of the users from the images. If an image

was upscaled, we assume it is of good quality and matches the user's intentions.

Although this is a reasonable assumption, this is not always the case. A user can upscale an image because they think the image is so bad that it is funny, or if they want to record the creation process. We expect it, however, to be of a small effect on the general "upscale" distribution.

Out of all the prompts, $25\%$ were upscaled.

## 4 Splitting into Threads

We split the prompts into threads. Each thread should contain a user's trails to create one target image. However, it is often difficult to determine whether the user had the same image in mind when they tried two consecutive prompts. For example, when a user asks for an image of "kids playing at the school yard" and then replaces "kids" with "a kid", it is hard to tell whether they moved to describe a new scene or only tried to change the composition. We consider a prompt to belong to a new thread according to the following guidelines:

1. Even when ignoring the subtle details, the current prompt describes a whole different scene than the previous one. It excludes cases where the user changed a large element in the scene, but the overall intention of the scene was not altered.

2. The main subjects described in the current prompt are intrinsically different from the subjects in the previous one. For example, an intrinsic change would be if in the previous prompt the main character was a cat, and in the current it is a dinosaur. If a kid is changed into kids, or a boy is changed into girl, it is not. An expectation is when a non-intrinsic change seems to change the whole meaning of the scene.

3. The current prompt can not be seen as an updated version of the previous prompt.

More examples with explanations are provided in §A.

### 4.1 Automatic Thread Splits

We propose methods to split the prompts into threads. $7,831$ of the users have one prompt only, so we mark each of them as an independent thread. To handle the remaining prompts, we use the following methods:

---

[3]We use spacy language detector https://spacy.io/universe/project/spacy_cld

[4]See full list here: https://docs.midjourney.com/docs/parameter-list

**Intersection over Union.** For each pair of consecutive prompts, we compute the ratio between their intersection and union. If one sentence is a sub-sentence of the other sentence, or the intersection over union is larger than 0.3, we consider the sentences to be in the same thread. Otherwise, we set the second prompt to be the first prompt of a new thread.

**BERTScore.** For each pair of consecutive prompts, we compute the BERTScore similarity (Zhang et al., 2019). If the BERTScore is larger than a threshold of 0.9, we put the sentences in the same thread.

We note that both methods assume non-overlapping threads and do not handle interleaved threads where the user tries to create two (or more) different scenes simultaneously.

## 4.2 Human Annotation Evaluation

We annotate prompts to assess the validity of the automatic thread splitting methods. We sampled users with at least 4 prompts and annotated their prompts. In this way, we increase the probability of annotating longer threads. We use the principles from §4 to manually annotate the prompts. One of the paper's authors annotated 500 prompts, and two more authors re-annotated 70 overlapping prompts each to assess inter-annotator agreement. While annotating the prompts, we found only 7 cases of interleaved threads (§4.1). We convert them to separate threads to allow the use of metrics for quality of linear text segmentation.

The agreement level between the three annotators was high (0.815), measured by Fleiss' kappa. Comparing the intersection over union annotations to the 500 manual annotations, we get an F1 score of 0.87 and average WindowDiff (Pevzner and Hearst, 2002) of 0.24 (the lower the better). For the BERTScore annotations, we get an F1 of 0.84 and average WindowDiff 0.30. Finding the intersection over union to be better, we select it to create the threads that we use for the rest of the paper.

## 4.3 Threads Statistics

With our automatic annotation method we get $107,051$ threads. The average length of a thread is $1.58$ prompts, with std $1.54$. See Figure 2. Each user produced $3.52$ different threads on average with std $12.67$. The longest thread is of length 77.

The average number of prompts that were up-scaled for each thread is $0.4$ with std $0.82$.

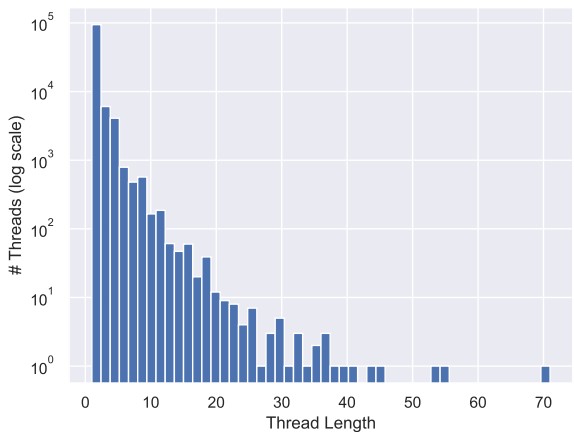

Figure 2: Histogram of the threads' lengths. There are a total of $107,051$ threads (annotated automatically), $645$ (.6%) of them contain 10 prompts or more.

## 5 Method

Our end goal is to examine the evolution of the prompts through the interaction. We start, however, by asking a simpler question, namely whether there is a difference between the upscale and non-upscale prompts. Such a difference would indicate that there are predictable characteristics (however intricate) of a prompt that render it better for Midjourney, and therefore provide motivation for the human users to adapt their prompts towards it.

To highlight differences between the upscaled and non-upscaled, we compile a list of linguistic features, that are potentially relevant to the up-scale decision. We find several features that are statistically different between the upscaled and non-upscaled populations, and then use those features to test the evolution of the threads.

We stress that we do not argue that these features account for a large proportion of the variance *between* users. Indeed, people use Midjourney for a wide range of tasks, with different levels of experience, hence their prompts and preferences vary a lot. Instead, we wish to make a principle point that there is a systematic convergence along the threads, and that it has practical implications. Future work will control for the intentions of the user, in which case we expect the convergence to account for a larger proportion of the variance.

## 5.1 Image and Text Classifiers

There are evidence that predicting whether the user was satisfied with the resulting image is possible given the prompt and image (Hessel et al., 2021;

Kirstain et al., 2023).

We hypothesize that the generated image alone would still allow a good guess, looking at the general quality of the image. More surprising would be to predict the upscale decision of the user based on the prompt alone. For that, there has to be a special language style or content type that leads to good images.

We formalize it as a partial input problem (Gururangan et al., 2018; Feng et al., 2019; Don-Yehiya et al., 2022) – predicting whether a prompt and image pair was upscaled or not, based on the image alone, or the prompt alone. We do not expect high performance, as this is both partial input and noisy.

We split the dataset to train and test sets (80/20), and sample from both an equal number of upscaled and non-upscaled prompts to balance the data. We finetune both a Resnet18 (He et al., 2015) and a GPT-2 (Radford et al., 2019) with a classification head on top of it (see §B for the full training details). The input to the model is an image or prompt respectively, and the output is compared to the gold upscaled/non-upscaled notion.

## 5.2 Linguistic Features Analysis

The classification model acts as a black box, withholding the features it uses for the classification. We hence compile a list of linguistic features that may be relevant to the upscale decision (Guo et al., 2023). For each of the features, we use the Mann–Whitney U test (Nachar, 2008) to examine whether the upscaled and non-upscaled prompts are from the same distribution or not. In App. D we apply this method also to the *captions* of the generated images, to examine the semantic properties of the generated images.

**Prompt Length.** We compare the length of the prompts in words.

**Magic Words.** We use the term "magic words" to describe words that do not add any real content to the prompt, but are commonly used by practitioners. For example, words like "beautiful", "8K" and "highly detailed." They all appear more than 1000 times in the dataset, but it is not clear what additional information they add to the scene they describe. Their popularity is due to the online community, claiming that the aesthetics and attractiveness of images can be improved by adding certain keywords and key phrases to the textual input prompts (Oppenlaender, 2022).

We identify 175 words that are probable in our dataset but not in general (see App. E).

For each prompt, we count the number of magic words in it, and normalize it by the prompt length to obtain the magic words *ratio* $\frac{\#magic\_words}{\#words}$.

**Perplexity.** We compute the perplexity GPT-2 (Radford et al., 2019) assigns to each prompt. We use the code from the Huggingface guide. A prompt with lower perplexity is a prompt that the model found to be more likely.

**Concreteness.** We use the concreteness ratings from Brysbaert et al. (2013) to assign each word with a concreteness score ranging from 1 (abstract) to 5 (concrete). We average the scores of all the words in the prompt to get a prompt-level score.

**Repeated Words.** For each prompt, we count the occurrences of each word that appears more than once in the prompt, excluding stop words. We then normalize it by the length of the prompt.

**Sentence Rate.** We split each prompt to its component sentences according to the spacy parser. We divide the number of words in the prompt by the number of sentences, to get the mean number of words per sentence.

**Syntactic Tree Depth.** We extract a constituency parse tree of the prompts with the Berkeley Neural Parser (Kitaev and Klein, 2018). We take the depth of the tree as an indication of the syntactic complexity of the sentence.

## 5.3 Analysis of Thread Dynamics

In the previous sections (§5.1, §5.2), we examined the end-point, namely whether the prompt was upscaled or not. In this section, we characterize the dynamics of the prompts along the thread, to identify the learning process undertaken by human users. For each feature that changes between the upscaled and non-upscaled prompts (§5.2), we are looking to see whether these features have a clear trend along the interaction, i.e., whether they are approximately monotonous. We plot the feature's average value as a function of the index of the prompt in the thread. We filter threads with less than 10 prompts, so the number of prompts averaged at each index (from 1 to 10) remains fixed.[5] This

---

[5]Without this restriction, we are at risk of getting mixed signals. For example, if we see that the first prompts are shorter, we can not tell whether this is because the shorter threads contain shorter prompts or because the threads are getting longer along the interaction.

| | Length | Magic | Perplexity | Concreteness | Repeated | Sent Rate | Depth |
|---|---|---|---|---|---|---|---|
| Upscaled | 16.67 | 0.109 | 2173 | 3.2628 | 0.040 | 14.19 | 6.19 |
| Not | 14.78 | 0.096 | 2855 | 3.2629 | 0.035 | 12.63 | 6.00 |
| $p$ value | $\mathbf{1.2e^{-231}}$ | $\mathbf{8.6e^{-80}}$ | $\mathbf{5.3e^{-80}}$ | 0.123 | $\mathbf{3.4e^{-56}}$ | $\mathbf{2.4e^{-191}}$ | $\mathbf{1.8e^{-49}}$ |

Table 1: The linguistic features values for the upscaled and non-upscaled prompts, with their matching $p$ value. All the features excluding the concreteness found to be significant. Magic words ratio, perplexity and repeated words ratio may indicate more 'model like' language, while the length, depth and sentence rate may indicate more details.

leaves 645 threads. In §G we present results for longer iterations (20), without filtering the shorter threads.

# 6 Results

In the following sections, we apply the method described in §5 to test the distribution and the dynamics of the prompts and threads. We start by identifying that there is a difference between the upscaled prompts and the rest (§6.1). We then associate it with specific linguistic features (§6.2). Finally, we examine not only the final decision, but the dynamics of the full interaction (§6.3).

## 6.1 Upscaled and Non-Upscaled are Different

We train and test the image classifier on the generated images as described in §5.1. We get an accuracy of $55.6\%$ with std $0.21$ over 3 seeds. This is 5.6 points above random as our data is balanced.

We do the same with the text classifier, training and testing it on the prompts. We get an accuracy of $58.2\%$ with std $0.26$ over 3 seeds. This is 8.2 points above random.

Although this accuracy is not good enough for practical use cases, it is meaningful. Despite the noisy data and the individual intentions and preferences of the users (that we do not account for), it is possible to distinguish upscaled from non-upscaled images/prompts at least to some extent.

We conclude that both the prompts and the generated images are indicative to the upscale question.

## 6.2 Significant Features

In the previous section we found that the upscaled prompts can be separated from the non-upscaled prompts to some extent. Here, we study what specific linguistic features correlate with this distinction, to be able to explain the difference.

Table 1 shows the mean values of the upscaled/non-upscaled prompts and the $p$ values associated with the Mann–Whitney U test. All the features except the concreteness score were found to be significant after Bonferroni correction

($p < 0.0007$), indicating that they correlate with the decision to upscale.

## 6.3 Thread Dynamics

To examine the dynamics of the threads, we plot in Figure 3 the significant features as a function of the index of the prompt. We see clear trends: the features are approximately monotonous, with some hallucinations. The length, magic words ratio, repeated words ratio and the sentence rate go up, and the perplexity down.

The magic words ratio has more hallucinations than the others, with a drop at the beginning. Also, unlike the other features, it does not seem to arrive to saturation within the 10 prompts window.

These results suggest that the users are not just randomly trying different prompt variations until they chance upon good ones. Instead the dynamics is guided in a certain direction by the feedback from the model. Users then seem to adapt to the model, without necessarily noticing.

# 7 Driving Forces Behind the Dynamics

We examine two possible non-contradictory explanations to the characteristics of the upscaled and non-upscaled prompts, and to the direction in which the human users go along the interaction. We show supporting evidence for both.

**Option #1: Adding Omitted Details.** When users input a prompt to Midjourney and receive an image, they may realize that their original prompt lacked some important details or did not express them well enough. Writing a description for a drawing is not a task most people are accustomed to. Hence, it makes sense that users learn how to make their descriptions more accurate and complete along the interaction.

Results from three features support this explanation. The prompt **length**, the **sentence rate** and the **syntactic tree depth**, all of them increase as the interaction progresses. Improving the accuracy of a prompt can be done by adding more words to describe the details, thus extending the prompt

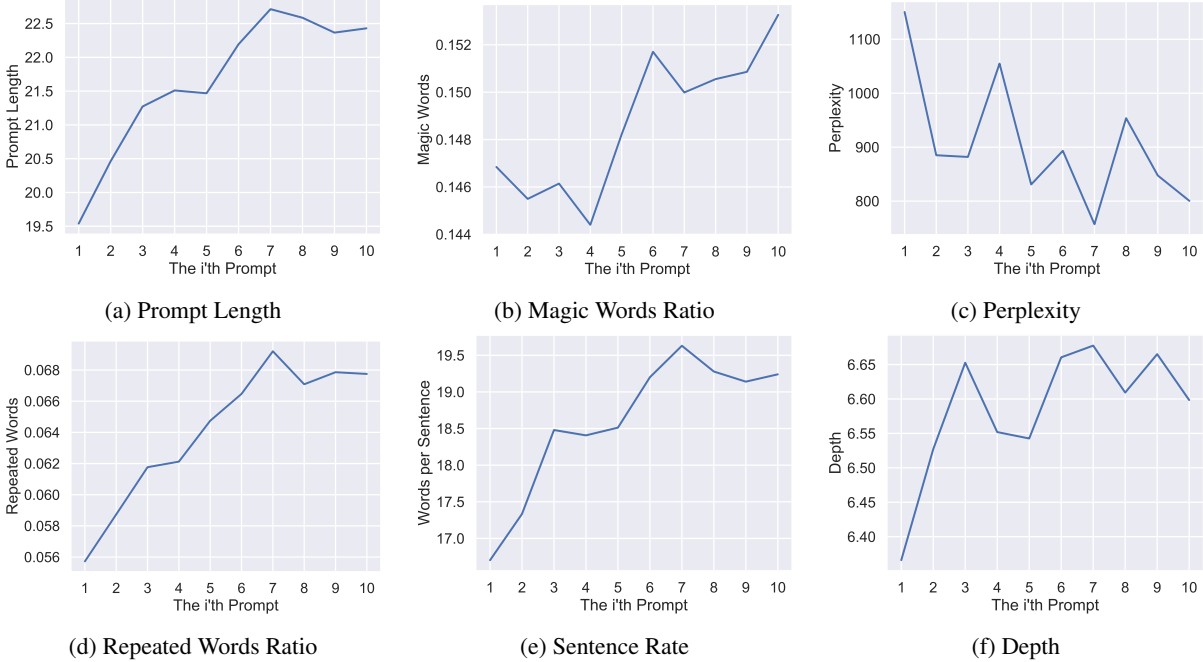

(a) Prompt Length      (b) Magic Words Ratio      (c) Perplexity

(d) Repeated Words Ratio      (e) Sentence Rate      (f) Depth

Figure 3: Average value for each of the significant linguistic features (y-axis), as a function of the prompt index (x-axis). The features are approximately monotonous, with some hallucinations. Most of the features go up (length, magic words ratio, repeated words ratio, sentence rate and tree depth) and the perplexity down. The users are not randomly trying different prompts until they reach good ones by chance, they are guided in a certain direction.

length. The new details can also increase the overall complexity as reflected in the number of words per sentence and the depth of the tree.

Another relevant feature is the concreteness score, as one can turn a sentence to be more accurate by changing the existing words to more concrete ones rather than adding new ones. Our results, however, show that the difference between the concreteness scores is not statistically significant.

**Option #2: Adopting Model-Like Language.** Another possible explanation is that human users learn to write their prompt in a language that is easier for the Midjourney model to handle. Human users try to maximize good images by adapting to "the language of the model".

Results from several features support this explanation. The **magic words** ratio is one of them. As mentioned in §5.2, magic words do not add new content to the prompt, and from an information-theoretic standpoint are therefore mostly redundant. Yet, there are more magic words in the prompts as the interaction progresses (even when correcting for the prompt's increasing length), suggesting that this is a preference of the model that the human users adapt to.

Another such feature is **perplexity**. The lower the perplexity the higher the probability the language model assigns to the text. The perplexity of the upscaled prompts is lower than the perplexity of the non-upscaled ones, and so is the perplexity of the 10th prompts compared to the first prompts. The users adapt to prompts that the model finds to be more likely. We note that it is possible to associate the descent in perplexity with the rise in length. While not a logical necessity, it is common that longer texts have lower perplexity (Lu et al., 2022).

Another feature that indicates human adaptation is the **repeated words** ratio. Repeated words usually do not add new information to the content, and therefore using them is not efficient. Our results, however, indicate that human users do it more often as the interaction with the model progresses. It is possible that they are used to simplify ideas for the model, or to push it to give more attention to certain details.

# 8 Convergence Patterns

In the previous section, we provided two explanations for the observed adaptation process. In this section, we further investigate the direction and destination of the adaptation.

For each feature $f$ we split the threads into two

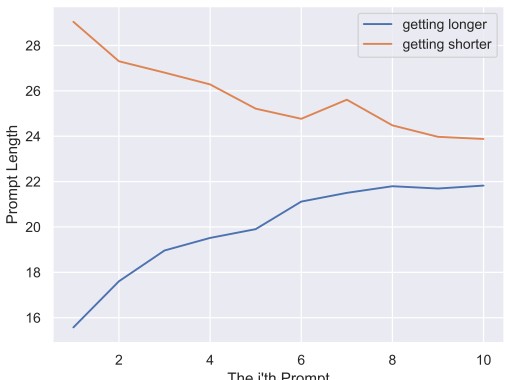

Figure 4: The threads that get longer start relatively short, and the threads that get shorter start relatively long. Both thread groups converge towards the same length range.

sets by their first and last values:

$$S_1 := \{\, thread \mid \text{f}(thread[0]) < \text{f}(thread[-1]) \,\}$$
$$S_2 := \{\, thread \mid \text{f}(thread[0]) \geq \text{f}(thread[-1]) \,\}$$

For example, for the length feature we have one set with threads that get longer, i.e., their last prompt is longer than their first prompt. In the other set, we have threads that are getting shorter, their last prompt is shorter than their first prompt. We see in Figure 4 that the prompts that get longer start shorter, and that the prompts that get shorter start longer:

$$\underset{thread \in S_1}{mean}\, f(thread[0]) < \underset{thread \in S_2}{mean}\, f(thread[0])$$

Both sets converge towards similar lengths. It may suggest that there is a specific range of "good" prompt lengths, not related to the starting point, and that human users converge to it, increasing or decreasing the length of the prompt adaptively depending on where they started. We observe similar trends in some of the other features too (see App. §H).

## 9 Discussion

In Section 7, we examined two explanations for the observed systematic adaptation process. The second option, that the model causes users to "drift" towards its preferences, raises concerns about naïvely using human data for training. Human data (compared to synthetic data (Honovich et al., 2022; Wang et al., 2023)) is often perceived as most suitable for training. The Midjourney data can be used

as RLHF data, sampling from each thread one image that was upscaled and one that was not, coupled with the upscaled prompt. However, our findings that the human users likely adapt to the model call for caution, as we may inadvertently push models by uncritically using user data to prefer the adapted prompts even more.

We hope to draw attention to the linguistic adaptation process human users go through when they interact with a model. Future work will empirically examine the effect of training with such data, and will expand the discussion on interactions with general language models.

## 10 Related Work

Text-to-image prompt engineering was studied by a handful of works. Oppenlaender (2022) identified prompt patterns used by the community, Pavlichenko and Ustalov (2022) examined the effect of specific keywords, and Lovering and Pavlick (2023) studied at the effect of subject-verb-object frequencies on the generated image.

Other works try to improve prompts by creating design guidelines (Liu and Chilton, 2022), automatically optimizing the prompts (Hao et al., 2022) or suggest prompt ideas to the user (Brade et al., 2023; Mishra et al., 2023).

The closest to ours is Xie et al. (2023), an analysis of large-scale prompt logs collected from multiple text-to-image systems. They do refer to "prompt sessions", which they identify with a 30-minute timeout. However, they do not split the prompts by scene, nor examine the dynamics of certain linguistic features changing along the interaction.

DiffusionDB (Wang et al., 2022) is a text-to-image dataset, containing prompts and images generated by Stable Diffusion (Rombach et al., 2021). It does not contain indications as to whether the user upscaled the image or not, and is therefore not suitable for our purposes. Another existing text-to-image dataset is Simulacra Aesthetic Captions (Pressman et al., 2022), containing prompts, images and ratings. It does not contain meta-data such as user-id or timestamps which make it unsuitable for our purposes. Another resource is the Midjourney 2022 data from Kaggle.[6] It contains raw data from the Midjourney Discord between June 20 and July 17, 2022. We did not use it but

---

[6]https://www.kaggle.com/datasets/ldmtwo/midjourney-250k-csv

scraped the data by ourselves, to obtain a larger and more recent dataset of prompts and of a newer version of Midjourney.

## Limitations

Our results explain a relatively small proportion of the variance between the upscaled and non-upscaled prompts. Similarly, the effects we show are statistically significant, presenting a conceptual point, but not large. Users have different levels of experience and preferences, and therefore their prompts and decision to upscale an image diverge. Future work will control for the content and quality of the prompts, in which case we expect to be able to explain a larger proportion of the variance.

We suggest two possible explanations regarding the observed convergence. We do not decide between them or quantify their effect. We do however show evidence supporting both, implying that both possibilities play a role.

As mentioned in §4.3, most of the threads are short, with one to two prompts. This is not surprising, as not all the users spend time in improving their first prompt. That left us with 6578 threads of at least 4 prompts each, 2485 of at least 6, and 1214 of at least 8. This may not be sufficient for future analyses. We will share the code we used to collect and process this dataset upon publication, so it will be always possible to expand this more.

During our work on this paper, a newer version of Midjourney was released (v5). It is very likely that the updates to the model would affect the prompts too, and all the more so if we will analyze prompts from a completely different system (we do successfully reproduce the results with Stable Diffusion data, see §F). However, we are not interested in specific values and "recipes" for good prompts. We only wish to point out the existence of adaptation and convergence processes.

## Ethics Statement

We fully anonymize the data by removing user names and other user-specific meta-data. Upon publication, users will also have the option to remove their prompts from our dataset through an email. Our manual sample did not find any offensive content in the prompts. The Midjourney Discord is an open community which allows others to use images and prompts whenever they are posted in a public setting. Paying users do own all assets they create, and therefore we do not include the image files in our dataset, but only links to them.

## Acknowledgements

This work was supported in part by the Israel Science Foundation (grant no. 2424/21).

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

## A Thread Examples

We provide threads examples with explanations. The full data is available at https://github.com/shachardon/Mid-Journey-to-alignment.

The following is an example for a 4 prompts length thread.

1. walking tiger from side simple vector clean lines logo 2d royal luxurious 4k

2. walking tiger from side simple vector clean lines logo 2d royal luxurious 4k in white background

3. running tiger from side simple vector clean lines logo 2d royal luxurious 4k white background png

4. jumping running tiger with open mouth from side simple vector clean lines logo 2d royal luxurious 4k gold and black with white background

The prompts describe the same main subject and scene, only small details are applied.

The following prompts are not similar to each other as the prompts from the previous thread are, but they do belong to one thread:

1. cloaked man standing on a cliff looking at a nebula

2. destiny hunter, standing on a cliff, looking at blue and black star

3. cloaked hunter, standing on a cliff, looking at a blue and black planet

The following two prompts are of the same user, created one after another, but are not part of one thread:

1. The girl who is looking at the sky as it rains in the gray sky

And -

| | Length | Magic | Perplexity | Concreteness | Repeated | Sent Rate | Depth |
|---|---|---|---|---|---|---|---|
| Upscaled | 15.52 | 0.101 | 2403 | 3.2692 | 0.035 | 13.15 | 6.13 |
| Not | 13.96 | 0.091 | 3081 | 3.2666 | 0.0313 | 11.94 | 5.951 |
| $p$ value | $\mathbf{1.0e^{-217}}$ | $\mathbf{1.5e^{-54}}$ | $\mathbf{1.2e^{-67}}$ | 0.021 | $\mathbf{3.3e^{-40}}$ | $\mathbf{1.3e^{-145}}$ | $\mathbf{6.6e^{-37}}$ |

Table 2: We rerun the experiment from §5.2, this time with prompts with default parameters only. All effects from the original non-filtered experiment persist.

1. Rain in the gray sky, look at the sky, Bavarian with a sword on his back

Although the scene is similar (rain, gray sky, a figure it looking at the sky), the main subject is intrinsically different (a girl / a male Bavarian with a sword).

The following three prompts constitute a thread:

1. the flash run

2. the flash, ezra miller, speed force

3. the flash running through the speed force

But the next prompt -

1. superman henry cavill vs the flash ezra miller movie

Is starting a new one, as both the main subjects (flash *and* superman) and scene (not running) are different.

The following prompts are not part of one thread:

1. A man rides a flying cat and swims on a snowy mountain

And -

1. A flying cat eats canned fish

The main subject is similar, but the scene is different.

## B Classifier Training Details

For the image classification we use a Resnet18 model (11M parameters) (He et al., 2015) that was pretrained on ImageNet (Russakovsky et al., 2015). We use batch size 8, SGD optimizer, learning rate 0.001, momentum 0.9, and X epochs. The input to the model is the 4 images grid. We tried to take as input the first image only, but it degraded the results.

For the text classifier model, we use GPT-2 (117M parameters) (Radford et al., 2019) with a classification head on top of it. The also experimented with RoBERTa-large (Liu et al., 2019) and

DeBERTa-large (He et al., 2023). We use batch size 16, AdamW optimizer, learning rate $2e - 5$, weight decay 0.01, and 3 epochs.

We train each instance of the models on 2 CPU and 1 GPU. The image classifier took the longest to train, about 20 hours, probably due to the time it takes to load the images from their url links.

We did not perform a hyperparameters search, as we only wanted to state a conceptual claim.

## C Default Parameters Only

It is possible that predefined parameters (see §3.1) have an effect on output acceptability. Therefore, we rerun the experiment from §5.2, this time with prompts with default parameters only. Filtering out prompts with any predefined parameters (i.e. non default parameters) leaves us with $147,236$ prompts. In table 2 we see that the results are similar to those of the non-filtered experiment, with all the effects from the original experiment preserved.

## D Applying our Method to the Captions

So far, we used our methodology to examine the properties of the prompts between upscale and non-upscaled versions and along the interactions. We now examine whether our conceptual framework can be used also for inferring the semantic properties of the generated images.

We already have indication that the upscaled and non-upscaled images can be distinguished from each other (§6.1). However, using the images themselves for the classification, we cannot separate the aesthetics of the image (e.g., whether the people's faces look realistic or not) from its content (e.g., what characters are in the image, what are they doing), nor to examine the linguistic features we already found to be relevant for the prompts.

For that, we represent each image with a textual caption that describes it. To perform the analysis at scale, we use automatically generated captions.

We use BLIP-2 (Li et al., 2023) to generate caption for the first image (out of the grid of 4 images) associated with a given prompt. We extract the same linguistics features we did for the prompts

| | Length | Magic | Perplexity | Concreteness | Repeated | Sent Rate | Depth |
|---|---|---|---|---|---|---|---|
| Upscaled | 9.292 | 0.048 | 356 | 3.095 | 0.013 | 9.290 | 5.832 |
| Not | 9.233 | 0.049 | 455 | 3.080 | 0.016 | 9.228 | 5.788 |
| $p$ value | $8.8e^{-7}$ | **0.0002** | $1.7e^{-12}$ | $9.2e^{-29}$ | $1.1e^{-8}$ | $4.3e^{-7}$ | $4.4e^{-8}$ |

Table 3: The linguistic features values for the captions that match the upscaled and non-upscaled prompts. All the features are significant, including the concreteness score which was not significant for the prompts themselves. Compared to the prompts, the captions are shorter, have lower magic words ratio, lower perplexity, shorter sentences and smaller effects size. The magic words ratio and the repeated words ratio are lower for the non-upscaled captions, the opposite of the prompts case.

(§5.2), and use the Mann–Whitney U test on them. In Table 3 we see that all the features were found to be significant, even the concreteness score that was not significant for the prompts. However, the effect sizes are smaller, and the direction of some of the effects is different. The magic words ratio and the repeated words ratio are both *lower* for the upscaled images compared to the non-upscaled, instead of higher like it was for the prompts.

We can speculate that the smaller effects are due to the image captioning model, which was trained to generate captions in a relatively fixed length and style, similar to those of the training examples it was trained on. The length, tree-depth and the sentence rate seems consistent with what we saw for the prompts. It is a little odd, however, as while a human user can choose to mention or not the color of the shoes that the kid in the scene is wearing, the shoes will have a color anyway, and the caption model would presumably handle it the same way.

A possible explanation is that there is more content in the upscaled images. For example, a dragon and a kid instead of a dragon only. Another option is that the details in the upscaled images are more noteworthy. For example, if the kid has blue hair and not brown. As for the magic word ratio and the repeated words ratio, they may strengthen the hypothesis that their rise in the prompts is a result of adaptation to the model's preferences; indeed, we do not see a similar effect in the captions. We defer a deeper investigation of this issue to further research.

## E Magic Words List

To find the magic words, we count the number of appearances of each word in the whole dataset, and normalize it by the total number of words to obtain a probability $P_{midj}(word)$. We then query google-ngrams[7] to get a notion of the general probability of each word $P_{general}(word)$, and

---

[7] https://books.google.com/ngrams/

divide the probability of a word to appear in a prompt by its "regular" google appearance probability $\frac{P_{midj}}{P_{general}}(word)$. We say a word is a "magic word" if it appears at least 1000 times at the dataset $P_{midj}(word) > 1000$ and the probability ratio is at least $\frac{P_{midj}}{P_{general}}(word) > 100$.

The full magic words list (total 175 words):

realistic, style, logo, background, detailed, 8k, Lighting, 4k, ultra, cute, lighting, cinematic, hyper, colors, photo, cartoon, Ray, anime, 3d, intricate, photorealistic, photography, super, Cinematic, Reflections, illustration, render, futuristic, Tracing, Illumination, dinosaur, portrait, fantasy, dino, cyberpunk, neon, minimalist, Photography, mini, 8K, Screen, Grading, HD, colorful, Unreal, Engine, hyperrealistic, RTX, hd, 4K, Color, octane, Beautiful, Volumetric, SSAO, unreal, Depth, RGB, realism, volumetric, Shaders, poster, Realistic, TXAA, CGI, Studio, minimalistic, 32k, beautifully, FKAA, Traced, VFX, Tone, DOF, SFX, Ambient, Logo, tattoo, vibrant, Hyper, Soft, Lumen, Accent, VR, Mapping, AntiAliasing, Megapixel, Shadows, Occlusion, hyperdetailed, Incandescent, HDR, Diffraction, Optics, Chromatic, Aberration, insanely, Scattering, Backlight, Lines, Moody, Shading, Rough, Optical, curly, SuperResolution, ui, tshirt, vintage, ProPhoto, ultradetailed, ar, Fiber, OpenGLShaders, Glowing, Scan, ultrarealistic, Ultra, v4, grading, Shimmering, ux, Tilt, PostProduction, Shot, Displacement, pixar, Cel, Editorial, GLSLShaders, Blur, wallpaper, hdr, Photoshoot, ContreJour, sticker, Angle, occlusion, pastel, graded, Massive, 16k, watercolor, coloring, retro.

## F DiffusionDB Results

We repeat some of the experiments with data from the DiffusionDB dataset (Wang et al., 2022), a text-to-image dataset of prompts and images generated by Stable Diffusion (Rombach et al., 2021). As discussed in §10, this dataset does not contain indications as to whether the user upscaled the image

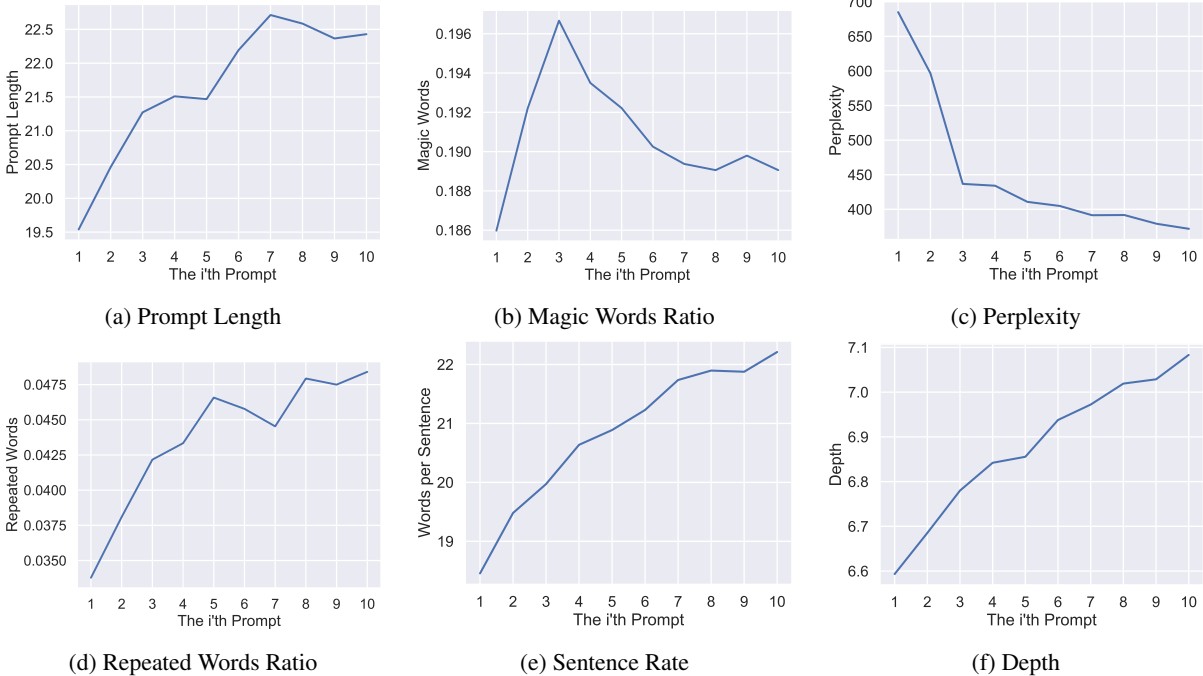

(a) Prompt Length      (b) Magic Words Ratio      (c) Perplexity

(d) Repeated Words Ratio      (e) Sentence Rate      (f) Depth

Figure 5: The thread dynamics experiment with the DiffusionDB dataset. Average value for each of the significant linguistic features (y-axis), as a function of the prompt index (x-axis). Except for the magic words ratio, all the features remain approximately monotonous like in our main experiment with the Midjourney dataset §3. Our features are relevant for the Stable Diffusion model too.

or not, and therefore we can not use it to reproduce the classifiers and the Mann–Whitney U test results (§6). We can however, use it to reproduce the thread dynamics experiment.

We take the first $250,000$ prompts of the 2M subset of the dataset. After cleaning (see §3.1), we remain with $105,644$ prompts. We split to threads (§4), resulting with $14,927$ threads, 1045 of them contain at least 10 prompts.

We present our results in §5. Except for the magic words ratio, all the features remain approximately monotonous like in our main experiment with the Midjourney dataset. Therefore, although not a logical necessity (see §10), it seems that most of our features are relevant for the Stable Diffusion model too.

## G   Longer Iterations

In §6.3, we restricted our analysis to threads with at least 10 prompts to avoid the risk of mixed signals. This restriction limits our ability to analyze longer threads, as there are few of them (there are only 67 threads with at least 20 prompts, see also Figure §2). Here, we loosen this restriction and use threads with at least 2 prompts, allowing the number of averaged prompts at each index to vary.

We double the number of iterations we are looking at. In Figure §6 we see that the features remain approximately monotonous.

We again note that this analysis is noisy. Not only that the later iterations average only few prompts, they are also possibly coming from a different distribution (the "long threads" distribution).

## H   Convergence Patterns for all the Features

Like we did in §6.3 with the prompt length, for each of the significant linguistic features we split the threads to two groups by the first and last values. One group contains threads that their first value is higher than their last value, and the second threads that their first value is lower than their last.

Like in the prompt length case, we can see in Figure 7 that the group that become longer start lower than the group that become shorter, and that they all go toward a narrower range.

This result is not trivial. For example, if we were to divide all the stocks in the stock market into those that rose during the day and those that fell, it is not true that those that rose started lower and those that fell started higher. If that were the case, we would have a clear investment strategy – buy

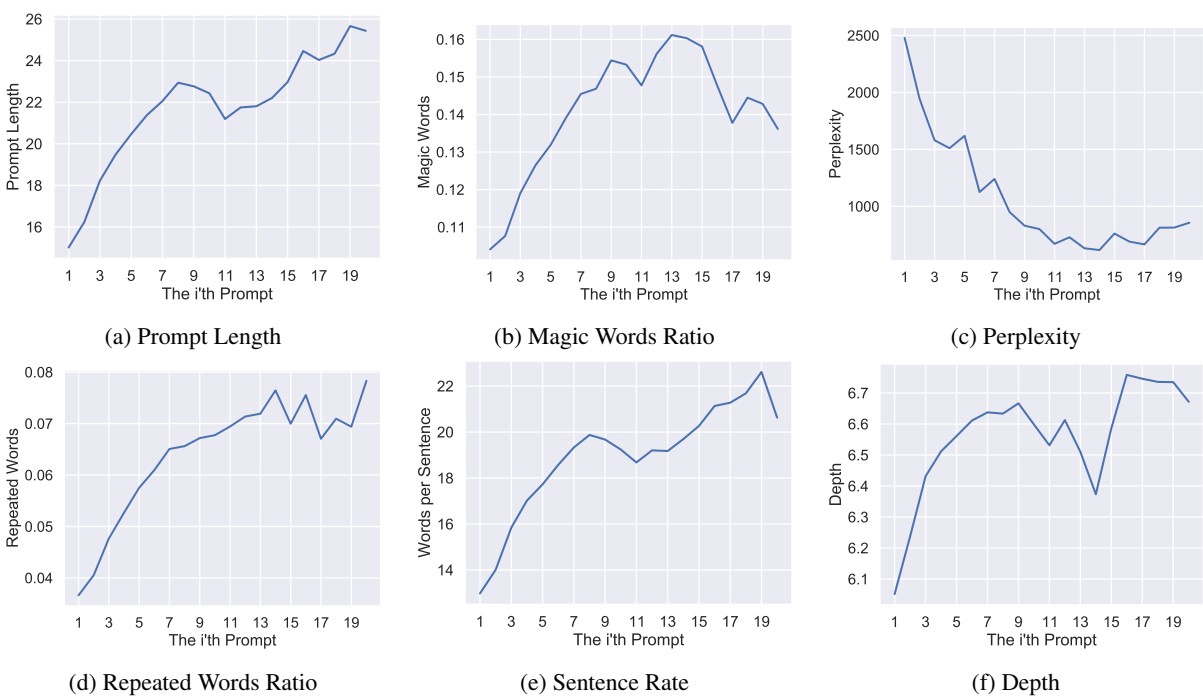

Figure 6: Average value for each of the significant linguistic features (y-axis), as a function of the prompt index (x-axis). We double the number of iterations we are looking at, and loosen our restriction, allowing the number of averaged prompts at each index to vary. The features remain approximately monotonous, with some hallucinations. Most of the features go up (length, magic words ratio, repeated words ratio, sentence rate and tree depth) and the perplexity down. The users are not randomly trying different prompts until they reach good ones by chance, they are guided in a certain direction.

only low-priced stocks.

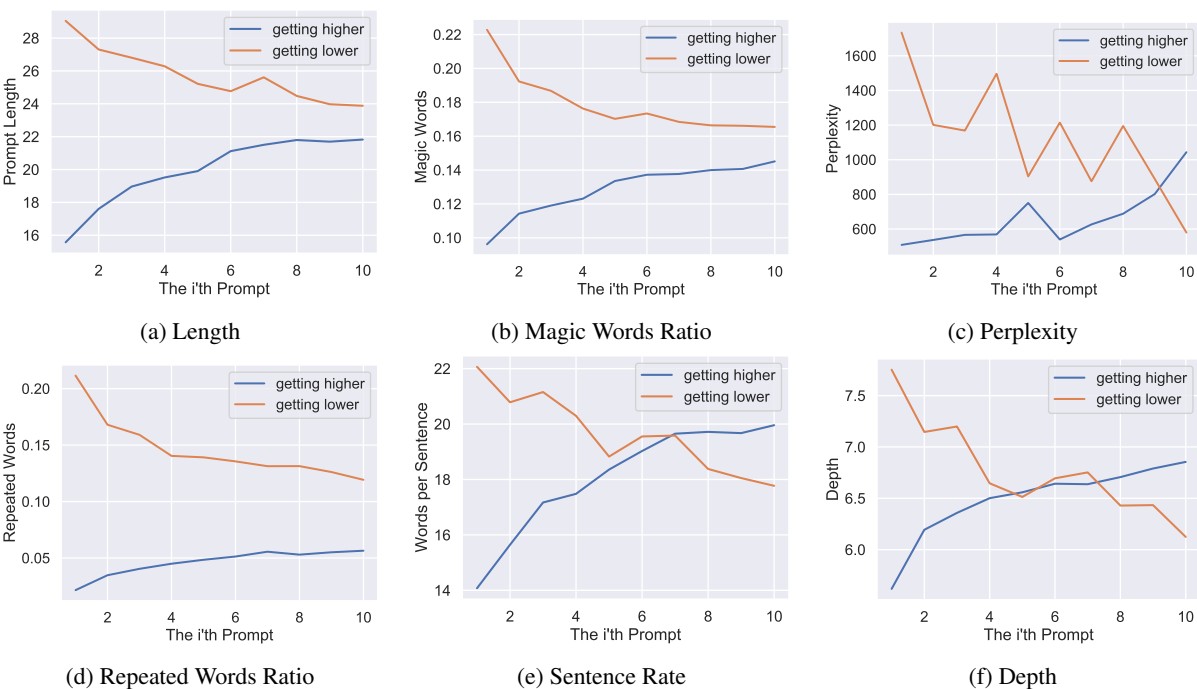

Figure 7: The groups that become longer start lower than the groups that become shorter, and they all go toward a narrower range, implying convergence to a specific "good" range.