# OpenReview forum: "Human Learning by Model Feedback: The Dynamics of Iterative Prompting with Midjourney"
_EMNLP/2023/Conference — EMNLP 2023 Main_

### Official Review · Reviewer_jjW3 · 2023-08-05

**Soundness:** 4

**Excitement:**

4: Strong: This paper deepens the understanding of some phenomenon or lowers the barriers to an existing research direction.

**Paper Topic And Main Contributions:**

This is a dataset and analysis paper that looks at human interactions with AI-assisted creative tools; specifically Midjourney for text-to-image generation. The authors find interesting results that there may be a feedback loop in play where humans are guided in their prompt design by model biases/preferences rather than their own goals. This is an interesting paper that studies a dynamic that is very relevant for societal ways of interacting with AI, and dives deep into the iterative dialog between users and the system. The authors ground their hypotheses in linguistic theory about the evolution/adaptation of language in discourse.

**Questions For The Authors:**

- What is the intuitive meaning of perplexity in 5.2? If GPT-2 found the prompt to be more likely, does this mean it's more likely to be a "human-like" sentence [judging from the training corpus which should pre-date midjourney] rather than a prompt written in a dedicated "midjourney style"?

**Reasons To Accept:**

- This analysis is highly topical and very relevant for the wider availability and common usage of AI models, as well as how users interact with these models via the modality of text
- The authors raise an important point about potential limitations when trying to harvest further datasets for model refinement - there is not necessarily an easy "free lunch" in bootstrapping a model from human interactions therewith.
- This dataset could have useful applications across a wide range of tasks, including: usage as a "contrastive" objective to force greater diversity in model responses, a method to refine midjourney prompts for a particular frozen text-to-image model version, gauging bias (and user sensitivity to bias) in image generations for prompts containing/omitting certain sensitive specifications (including race, color, gender, etc.).
- The paper is structured well and written clearly, both in terms of experimental design and result discussions.

**Reasons To Reject:**

- Would like to see greater discussion about the impact of removing predefined parameters (3.1) - this potentially has an effect on output acceptability (e.g. higher chaos or noise could lead to greater adjustment of prompts), and it would be good to see a version of analysis with for e.g. only prompts with the same hyperparameters.
- Why was 0.3 chosen as an IoU/Jaccard threshold, and why is BERTScore threshold of 0.9 used? It would be good to see illustrative examples at different thresholds to give an idea of how threads are split and what constituted a "new thread" in the analysis.

**Reproducibility:**

2: Would be hard pressed to reproduce the results. The contribution depends on data that are simply not available outside the author's institution or consortium; not enough details are provided.

**Reviewer Confidence:**

4: Quite sure. I tried to check the important points carefully. It's unlikely, though conceivable, that I missed something that should affect my ratings.

---

> ### Author Rebuttal · Authors · 2023-08-27
>
> We thank the reviewer for their support, as well as the clear and accurate summary.
>
> We will add in the camera ready experiments without any predefined parameters (so all set to default).
>
> The main figure contains two thread examples, we will be happy to add more examples to the paper, including some “negative” examples to show when a new thread starts.
> We chose the thresholds mostly manually – we tried different ones and looked at some prompts to decide whether it was better or not (prompts that were not included in the human annotation evaluation). Probably there are better choices, but as our human evaluation shows, they seem to be good enough to rely on for our analysis.
>
> Our intuition is that better perplexity means more model-like language, in contrast to human-like language.
> We repeated this experiment with GPT-2-medium and GPT-NEO, to vary the model size and the training data, and obtained the same trend. We therefore conclude that this preference is shared across many models, possibly Midjourney among them.
>
> The dataset and the code will be released upon acceptance; we hope this will facilitate reproducibility.

---

### Official Review · Reviewer_4xF9 · 2023-08-06

**Soundness:** 4

**Excitement:**

4: Strong: This paper deepens the understanding of some phenomenon or lowers the barriers to an existing research direction.

**Paper Topic And Main Contributions:**

This paper analyzed the statistic of linguistic features of the prompts used in the image generation with MidJourney. Two user behavior patterns are proposed and the extracted statistic supports the assumption about the dynamics.

**Reasons To Accept:**

- The detailed analysis of the prompts can facilitate understanding the user behavior during the image generation process.
- The two explanations are reasonable and well-supported by the linguistic feature statistic.

**Reasons To Reject:**

The explanation for adding omitted details is reasonable and convincing. This happens in many cross-modality tasks, such as image/video retrieval and generation. However, the evidence for the second point: Adopting Model-Like Language, is insufficient.
1. Users do not know whether a word is a magic word or not. If the knowledge of the magic word is from an external source, like a community, the interactions, it's hard to say they learn the way to use magic words. The increase in the magic words is likely to be a part of the complemented information by the users and has no difference from the other complemented words.
2. A learning process can be shown as the preference of a language style after trying different language styles. The change between different styles and the final convergence to a specific style can more sufficiently support the statement.

**Reproducibility:**

5: Could easily reproduce the results.

**Reviewer Confidence:**

3: Pretty sure, but there's a chance I missed something. Although I have a good feel for this area in general, I did not carefully check the paper's details, e.g., the math, experimental design, or novelty.

---

> ### Author Rebuttal · Authors · 2023-08-27
>
> Thank you for your positive review.
>
> We gave two “definitions” of magic words in the paper (section 5.2):
>
>
> * Words that do not add any real content to the prompt
> * Words that can improve the aesthetics and attractiveness of images, according to the online community
>
> When claiming that the increased use of magic words is an indication to the second explanation (model adaptation), we mainly focus on the first definition. We observe that users add to their prompts words that do not add substantial new information, something that can not be explained by the first explanation (adding missed details).
>
> We are not sure we fully understand the second point about language styles.  We interpreted it as asking human users to intentionally try several “styles” and then pick one rather than gradually levitate in a specific direction. This method will indeed be useful for finding what the model likes, but this is not our goal. We are interested in what the users are led to use organically. Please let us know if we did not understand correctly.

---

### Official Review · Reviewer_qxK5 · 2023-08-06

**Soundness:** 4

**Excitement:**

4: Strong: This paper deepens the understanding of some phenomenon or lowers the barriers to an existing research direction.

**Paper Topic And Main Contributions:**

This paper analyzes the dynamics of human prompts along iterative interactions with the Midjourney text-to-image model. The authors compile a dataset of 107K threads containing sequences of prompts and images generated by Midjourney. Their analysis reveals convergence patterns in prompt features like length, perplexity, and concreteness over the iterations. They suggest two explanations for this: (1) users adding omitted details, and (2) users adapting their language to match the model's preferences. They present evidence supporting both possibilities. The authors caution against reusing such data to further train models, as prompts become biased toward a particular model's preferences.

**Questions For The Authors:**

- What proportion of variance in prompting dynamics do the identified effects explain? How does controlling for user goals/experience affect this?
- How well do the results generalize to other models besides Midjourney? Are there common patterns?
- Can you further tease apart and quantify the relative impact of the two proposed explanation factors?
- How do the dynamics look over a larger number of iterations beyond 10? Is there continued convergence?
- How do your findings relate to human-AI conversations more broadly? Can similar effects be seen there?

**Reasons To Accept:**

1. Analyzes an important aspect of human-AI interaction dynamics that has received little prior attention. Understanding how users adapt their language when interacting with models over time provides useful insights.
2. Provides the new Midjourney iterative prompt dataset as a contribution for the research community. This data enables further analysis of human-model prompting dynamics.
3. Identifies several interesting patterns in how users change their prompts over iterations, such as increases in length, perplexity drops, and more concreteness. The trends indicate clear convergence effects.
4. Makes an important cautionary point about potential biases that could arise from reusing human-AI data. The prompts likely become adapted to a particular model's preferences, rather than natural human language.
5. The paper is generally clearly written and easy to follow. The introduction motivates the research question well. The method and analysis are adequately explained and presented.
6. The scope of focusing on textual prompting for a single model is reasonably narrow. This enables an in-depth analysis of prompting dynamics for Midjourney.
7. The analysis methodology using prompt feature analysis and thread dynamics seems appropriate and is able to uncover clear patterns in the data.

**Reasons To Reject:**

1. The identified effects account for a relatively small portion of variance between prompts. Individual user goals and experience likely play a large role that is not controlled for.
2. Only examines prompts from a single model, Midjourney. The results may not generalize well to other text-to-image models or conversational AI systems.
3. Provides evidence for the two proposed explanations, but does not further tease apart or quantify their relative impacts on prompting dynamics.
4. The results are over a limited set of 10 iterations. It is unclear if the patterns continue in the same direction over longer interactions.
5. The scope focusing solely on textual prompts for a single model is quite narrow. The findings do not necessarily apply to broader human-AI conversations.
6. While the identified patterns are interesting, they are not particularly surprising. The basic idea of users adaptively refining prompts matches intuitions.
7. The methodology analyzes overall dataset patterns. It does not look at differences in prompting dynamics across individual users or prompt topics.

**Reproducibility:**

3: Could reproduce the results with some difficulty. The settings of parameters are underspecified or subjectively determined; the training/evaluation data are not widely available.

**Reviewer Confidence:**

4: Quite sure. I tried to check the important points carefully. It's unlikely, though conceivable, that I missed something that should affect my ratings.

---

> ### Author Rebuttal · Authors · 2023-08-27
>
> We thank you for the deep reading and appreciate the attention to detail reflected in your review.
>
> * In this work we chose to look at threads in the wild, without the interference of instructing the annotators. We expect most of the variance to be explained by the users' goals and experience, which we do not control for. As mentioned in the limitations, we plan this for future work.
> Finding the proportion of the explained variance in these settings is not forthright in this case, as we do not expect to be able to predict the dynamics without these controls. Our most related finding to this in the paper is the text classifier results (section 6.1), which outperformed the random baseline by 8.2 accuracy points.
>
> * It is very likely that the patterns between different systems (and even between different versions of the same system) will vary.  As mentioned in the limitations section, we are not interested in specific values for good prompts. We only wish to point out the existence of adaptation and convergence in the threads. However, we will add some experiments with the StableDB dataset (restricted to the limited metadata it contains).
>
> * We are not sure how to tease apart and quantify the two explanations, but if you have any ideas in mind we would be very happy to try. We would like to emphasize that our suggested features are probably not the only ones, so although they are indeed useful in showing that these explanations are present, they might not be suitable for quantitative examination.
>
> * When looking at 10 iterations, we are limiting our analysis to threads of at least 10 prompts, to avoid mixed signals (maybe longer threads tend to contain longer prompts? We do have these results however, and we will add them to the appendix). Notably, those also form the vast majority of interactions. With this restriction, when looking at longer interactions we will have only a few threads to examine (see fig. 2). Note that in the significant features test (6.2) we compared the first and last prompts, so it does take longer iterations into account.
>
> * Handling other human-model conversations systems with the exact same method is challenging. Unlike text-to-image systems, other conversational systems tend to include multiple topics and vary along the conversation. We can not assume that the content remains relatively similar between the iterations as we did with the text-to-image model. We aim to find the right method for analyzing general conversational systems in future work.

---

### Meta-Review · Area_Chair_6v1Y · 2023-09-10

**Recommendation:** 4

**Metareview:**

All reviewers agree on the 4:4 scores for soundness and excitements and I can only agree with those reviews. The paper provides a novel and original contribution that the field can benefit from.

Pros:
- overall, the paper is clearly written, well structured and comprehensive
- the contribution of the dataset is valuable to the community
- idea and methodology are novel and very timely
- the limitations and critical assessment is appreciated

Cons:
- most of the criticism is raised with respect to inaccuracies within the paper which have been clarified in the rebuttal
- without the data/code available during review, the reproducibility is limited

Overall, I agree with the reviewers and agree that the paper is moderately sound and exciting. What is missing for 5:5 made apparent by the many (minor) ambiguities that raise the questions of all reviews.

---

### Decision · Program_Chairs · 2023-10-07

**Decision:**

Accept-Main

**Comment:**

All reviewers agree on the 4:4 scores for soundness and excitements and I can only agree with those reviews. The paper provides a novel and original contribution that the field can benefit from.

Pros:
- overall, the paper is clearly written, well structured and comprehensive
- the contribution of the dataset is valuable to the community
- idea and methodology are novel and very timely
- the limitations and critical assessment is appreciated

Cons:
- most of the criticism is raised with respect to inaccuracies within the paper which have been clarified in the rebuttal
- without the data/code available during review, the reproducibility is limited

Overall, I agree with the reviewers and agree that the paper is moderately sound and exciting. What is missing for 5:5 made apparent by the many (minor) ambiguities that raise the questions of all reviews.